# Flow Assisted Mutation Enrichment (FAME): A highly efficacious and efficient method to enrich Double Knockouts (DKO) after gene editing

**Michael Hansen[1], Xiaopin Cai[2¤], Sara Bowen[3], David A. Largaespada[2,4], Ming V. Li[1,5]***

**1** Division of Endocrinology, Phoenix VA Medical Center, Phoenix, Arizona, United States of America, **2** Department of Genetics, Cell Biology and Development, University of Minnesota, Minneapolis, Minnesota, United States of America, **3** Flow Cytometry Core Facility, Barrow Neurological Institute, Phoenix, Arizona, United States of America, **4** Department of Pediatrics, University of Minnesota, Minneapolis, Minnesota, United States of America, **5** Department of Medicine, University of Arizona College of Medicine, Phoenix, Arizona, United States of America

¤ Current address: Division of Endocrinology, China-Japan Friendship Hospital, Beijing, China
* Ming.Li@va.gov

**Data Availability Statement:** All relevant data are within the manuscript and its Supporting information files.

## Abstract

Gene editing has become an essential tool for interrogation of gene function in biomedical research and is also a promising approach for gene therapy. Despite recent progresses, the gene-editing procedure is still a tedious process involving manually isolating large number of single cell colonies to screen for desired mutations. For diploid eukaryotic cells, there is the additional challenge to inactivate both alleles for genes-of-interest, i.e., generating double knockouts (DKOs), for the desired phenotypes or therapeutic effects. In this report, we present a novel method based on Fluorescence Assisted Cell Sorting (FACS) to enrich for DKO cells, using a cell surface marker β2-microglobulin (B2M) as a basis for negative selection. This method significantly increased percentage of DKOs in isolated cells after gene editing, and in the meantime, significantly improve the efficiency of workflow by automating colony isolation. It would greatly facilitate future biomedical research including potential gene/cell therapies.

## Introduction

Recent progresses in designer nucleases have greatly improved the efficacy of gene editing, especially with the advent of Clustered Regularly-Interspaced Short Palindromic Repeats (CRISPR) technology [1]. Although CRISPR was originally discovered in prokaryotes, the system has since been developed into a gene-editing tool for eukaryotic cells as well [2]. The technology relies on short guide RNAs (gRNAs) to position Cas9 nuclease towards the target genomic loci to introduce double-stranded DNA breaks; mutations are introduced during the DNA repair process, with either non-homologous end joining, or more precisely with homologous recombination. The workflow of CRISPR gene editing has been described in detail by

**Funding:** This work was supported by the National Institutes of Health (Cancer.gov) [1R21CA201735-01 to D.A.L and M.V.L], and Arizona Veteran Research and Education Foundation Research Grant (https://carlthaydenmedicalresearchfoundation.org/)[to M. V.L]. The funders had no role in study design, data collection and analysis, decision to publish, or preparation of the manuscript.

**Competing interests:** The authors have read the journal's policy and the authors of this manuscript have the following competing interests: DAL is a co-founder and co-owner of NeoClone Biotechnologies, Inc., Discovery Genomics, Inc. (then acquired by Immusoft, Inc.), B-MoGen Biotechnologies, Inc. (then acquired by Bio-Techne Corporation), and Luminary Therapeutics, Inc. DAL holds equity in, serves as a Senior Scientific Advisor for, and on the Board of Directors for Recombinetics, a genome editing company. DAL consults for Genentech, Inc., which is funding some of his research. None of the work described in this manuscript is related to the business of these companies. A provisional patent application with M.V.L listed as inventor has been submitted by Office of Research and Development, Department of Veterans Affairs. The technology described in this manuscript has been filed for patent application under international filing # WO2020087010A1, application # PCT/US2019/058155 and title Compositions and methods for selecting biallelic gene editing. This does not alter our adherence to PLOS ONE policies on sharing data and materials.

Ran et al. [3]. Briefly, gRNAs against genes-of-interest are designed and cloned into targeting vectors for expression in the target cells; the constructs are then introduced along with Cas9 expressing plasmid into target cells. Successful target editing can be verified by Surveyor or T7 endonuclease I (T7EI) assays with genomic DNA extracted from edited cells, which takes advantage of nucleases specializing in cutting at mismatches in dsDNA (heteroduplex) as a result of indel mutations introduced by non-homologous end-joining (NHEJ) repair after gene editing. The edited cells are then separated into individual single cell colonies, with their individual mutations determined by PCR cloning of the targeted loci and Sanger sequencing. The entire process usually lasts about 4 weeks, with most of human hours spent in isolation of single cell colonies and genotyping them for desired mutations. The number of colonies to be screened could be significant, especially if one desires double knockout of the target genes in diploid eukaryotic cells, as most resulting colonies have only mutations to a single allele. A few attempts have been made to improve the success rate of creating DKOs. One strategy relies on sequential targeting of each allele of the gene-of-interest and inserting expression cassettes for different fluorescent proteins at the mutation sites by way of homology directed repair (HDR). This will allow cells with both alleles mutated to be positively selected with FACS [4]. Similarly, other HDR based targeting methods have been developed to insert positive selection markers including drug resistance gene and fluorescent markers to the targeted loci and have achieved varying degree of success [5, 6]. However, the above methods all involve complex workflow. For each gene being targeted, one needs to not only create a construct for sgRNA, but also repair templates with gene-specific arms for homologous recombination and different selection markers. For cells that are difficult to transfect, such strategy is unlikely to succeed.

Alternatively, a second strategy resorted to negative selection by concurrently targeting a separate locus that provides negative selection, along with a gene-of-interest. As an example, hypoxanthine-guanine phosphoribosyltransferase (HPRT) is an enzyme in the rescue pathway of purine synthesis. It is not essential for cell survival; yet its deficiency in host cells protects against cytotoxic drug 6-thioguannine (6TG). This feature was exploited to enrich for DKOs in nuclease-modified cells when both gene-of-interest and *HPRT* are co-targeted. The results are variable, with the percentage of DKO under 5% in successful runs [7]. The unsatisfactory results likely reflect the fact that *HPRT* is X-linked, therefore in any given sex, only one functional copy is in force. Therefore, 6TG resistant colonies result from modification of only a single allele, which does not provide adequate selection pressure for double knockouts. Alternatively, co-targeting $Na^+/K^+$ ATPase (encoded by gene *ATP1A1*) allows negative selection with ouabain and has significantly enriched mutation frequency for both indels and homology-directed repair [8]. However, as $Na^+/K^+$ ATPase plays a significant role in cell physiology, the functional implications to cells so targeted remain to be explored. In addition, isolation of both 6TG and ouabain resistant colonies requires manual labor and is time-consuming comparing to automated single cell isolation from FACS.

In this report, we describe an invention, hereafter referred to as Flow Assisted Mutation Enrichment (FAME), that combines the convenience of automated single cell isolation afforded by FACS and the power of negative selection, using a surface marker that provides adequate selection pressure. With this technology, we were able to significantly enrich for DKOs at the desired loci, which are co-targeted along with the negative selection marker. The FACS procedure in the workflow also provides the desired automation for single cell colony isolation. We produced evidence that FAME procedure has significantly increased the prevalence of indel mutations in the negatively selected cells, with close to 100% of isolated colonies being DKOs, a success rate previously only achieved with positive selection scheme, but with much more efficient workflow and wider applicability.

## Materials and methods

### Tissue culture

HEK293T cells were originally obtained from ATCC (Manassas, VA), and cultured in DMEM media with 10% FBS. Tissue culture media and supplies are provided by VWR (Radnor, PA), unless otherwise indicated.

### Plasmids and cloning

pCMV-hCas9, and pENTR221-U6-sgRNA constructs were kind gifts of Dr. Branden Moriarity of University of Minnesota. The sgRNA constructs for targeting *B2M*, *PTEN*, *MYC*, and *ZMIZ1* were created based on cloning of PCR products as described by Ran et al [3]. Briefly, for each construct, PCR was conducted using diluted pENTR221-U6-sgRNA as template, common reverse primer (5'-cggtgtttcgtcctttccac-3'), and guide-specific forward primers (Table 1) to create plasmid-length PCR products, which was then treated with T4 polynucleotide kinase (New England Biolab, Ipswich, MA) to enable self-ligation with T4 ligase (New England Biolab). The ligation products were used to transform competent DH10B *E. coli* (New England Biolab). Resulting plasmids were verified by Sanger sequencing using M13 reverse primer at DNA sequencing lab at Arizona State University (Phoenix, AZ). The target loci for editing of each gene by *Streptococcus pyogenes* Cas9 nuclease were chosen based on prediction by a cloud-based algorithm hosted at Deskgen.com (Desktop Genetics, London, UK). Among the candidates generated by the algorithm, we have chosen those with highest on-target and off-target score to achieve maximal editing efficacy and minimal off-target effects (Table 1).

### Transfection

Transfection of HEK293T cells is achieved with Lipofectamine 2000 (Invitrogen, Carlsbad, CA) according to the instructions of the manufacturer. In a typical co-targeting gene editing experiment with B2M negative selection, A total of 2.5 μg of plasmid DNA (1 μg pCMV-hCas9, 0.5 μg pENTR221-U6-sgB2M1, and 1 μg of plasmid for gRNA targeting the gene-of-interest) is used to transfect 1 well of HEK293T cells in a 6-well plate.

### Flow cytometry

Transfected HEK293T cells are stained with FITC-conjugated anti-Human HLA-A,B,C antibody (Clone W6/32, Part number B223308) (Biolegend, San Diego, CA) according to protocol provided by the manufacturer, and analyzed on a Beckman Coulter Cytomics FC500 flow cytometer. The results were analyzed with Beckman Coulter CXP Software (Brea, CA).

**Table 1. Sequence of oligos used for creating sgRNA expression plasmids.**

| Target | Sequence (5'-3') |
| --- | --- |
| *B2M #1* | CAGCCCAAGATAGTTAAGTGgttttagagctagaaatagc |
| *B2M #2* | ACAAAGTCACATGGTTCACAgttttagagctagaaatagc |
| *B2M #3* | CTGAATCTTTGGAGTACCTGgttttagagctagaaatagc |
| *PTEN* | ATGACCTAGCAACCTGACCAgttttagagctagaaatagc |
| *MYC* | CAGAGTAGTTATGGTAACTGgttttagagctagaaatagc |
| *ZMIZ1* | TTGGTTACTCCCCAAACCGgttttagagctaggccaac |

*Gene specific sequence is capitalized in contrast to common plasmid sequence

## FACS

Staining of 293T with anti-MHC-I antibody was achieved the same way as described above in flow cytometry analysis. Sorting of MHC-I negative and positive cells is carried out in a Becton Dickinson (Franklin Lakes, NJ) FACSAria IIu cell sorter in Barrow's Neurological Institute, Phoenix, AZ.

## T7 endonuclease I (T7EI) assay

The T7EI assay was conducted with kits purchased from New England Biolabs (Catalog # E3321S), according to manufacturer's instruction. Briefly, PCR was conducted with provided high-fidelity polymerase for each targeted locus with primers listed in Table 2. The PCR products were denatured and reannealed for heteroduplex formation, followed by digestion with T7 endonuclease I that recognizes mismatch created by mutagenesis. The digested products and undigested control were then run on 2% agarose gel to decide the presence and percentage of indel mutations as a result of CRISPR gene editing. The gel images were acquired with a Biorad Chemidoc XRS imager (Biorad, Hercules, CA). Band signal quantification was performed with Image Lab software pre-installed on the imager. Indel frequency was calculated below as previously described, i.e., Indel (%) = $(1-\sqrt{1-(b+c)/(a+b+c)}) \times 100$, where a is the integrated intensity of the undigested PCR product and b and c are the integrated intensity of theT7EI cleavage products [8].

## Statistical analysis

Comparison of indel frequency between unsorted and sorted cell population after co-targeting experiments against *PTEN*, *MYC* and *ZMIZ1* was performed with student's t-test using Microsoft Excel (Microsoft Inc, Redmond, WA). $P < 0.05$ is considered statistically significant.

## Sequencing verification of DKOs of *PTEN* in MHC-I negative and positive single cell colonies

Genomic DNA was extracted from expanded single cell colonies, and PCR was conducted to amplify the targeted *PTEN* locus with *PTEN* Fwd and Rev primers (Table 2). PCR products were column purified with PCR purification kit (Thermofisher Inc., Waltham, MA) and submitted for Sanger sequencing with *PTEN* rev primer (Table 2) at the DNA sequencing lab at Arizona State University. Clones with apparent wild-type sequences were identified as genetically unmodified. Those with mixed signals were further subject to cloning of the PCR product into pMini T2.0 vector with NEB PCR cloning kit (New England Biolab), unless DKO can be

**Table 2. Oligo sequences for T7EI assay for *PTEN*, *MYC* and *ZMIZ1*.**

| Target | | Sequence (5'-3') |
|---|---|---|
| *PTEN** | Fwd | CCAGGCCTCTGGCTGCTGAG |
| | Rev | CGGACAATAGCCCTCAGGAAGA |
| *MYC** | Fwd | CGGAGCGAATAGGGGGCTTC |
| | Rev | GGCCGGGAGTCAGCGTGAA |
| *ZMIZ1* | Fwd | CAGTTGCATGACCTGTGGAC |
| | Rev | GAAGCTGGTCTTTCCAGCAG |

* from reference [7].

called unequivocally from the chromatogram based on sequencing of the PCR products. The plasmids from PCR cloning of each sample were sent for Sanger sequencing with T7 primer at Arizona State University DNA sequencing lab to confirm presence of mutations in single or both alleles.

## Results

### Creating targeting plasmids for surface selection marker β2-microglobulin (B2M)

The success of a negative selection approach depends on an appropriate selection marker. We reasoned that an ideal marker needs to satisfy the following four conditions, i.e., 1) easily selectable; 2) autosomal thus biallelic, providing adequate selection pressure for successful gene editing; 3) supporting automation of single cell isolation, such as with FACS; and importantly 4) dispensable for survival and important cellular functions.

A membrane protein β2-microglobulin (B2M) could potentially satisfy all above conditions. B2M is a component of the type I Major Histocompatibility Complex (MHC-I), thus universally present in all tissue types. It can be readily detected with flow cytometry with antibodies against either B2M itself or the MHC-I complex [9]. Cells with surface ablation of B2M can thus be negatively selected with FACS. *B2M* gene is autosomal with 2 alleles located on chromosome 15; complete ablation of surface B2M protein requires loss-of-function mutations in both alleles, providing strong selection pressure for effective editing and allowing DKOs for the co-targeted genes to be enriched in the sorted cell population. The desired selection pressure can be manipulated with the ratio between targeting plasmids for the gene-of-interest and *B2M*.

Importantly, *B2M* and MHC-I are not essential for cell survival and largely dispensable in *ex vivo* settings. As part of MHC-I, B2M plays a role in the development and execution of cell-mediated immunity. Mice with *B2M* knockout are viable despite being immunodeficient for lack of CD8+ lymphocytes [10]. Cells lacking B2M are hypoimmunogenic and protected from cell-mediated immunity as they could not be recognized by CD8+ T lymphocytes. This feature can be taken advantage of to generate off-shelf and hypoimmunogenic cell therapy products originating from allogenic or xenogenic donors, thus greatly reducing the cost of cell therapy. In this setting, ablation of *B2M* can serve two purposes at the same time, including enriching for desired therapeutic genetic modifications, and offering protection against rejection of implanted cells. In an earlier proof-of-principle experiment, it was reported that human embryonic stem cells (hESC) with the surface B2M ablated were able to develop into teratoma once implanted into immunocompetent mouse, thereby showcasing both the negligible functional consequence of *B2M* ablation and its usefulness in developing cell therapy [11]. Ablation of *B2M* has also been instrumental in producing hypoimmunogenic CAR-T cells with little Graft-versus-Host Disease [12]. In addition, in patients with end-stage renal failure, elevated level of B2M is a source of amyloidosis, therefore cell therapy with ablated *B2M* would be highly desired in such situations as well [13].

We have designed three sgRNAs using cloud-based software from Desktop Genetics (London, UK) with predicted high targeting efficiency and low off-target editing (Table 1). We have tested the efficacy of all three with flow cytometry after transfecting them along with expression plasmid for spCas9. All three were able to produce significant portion of cells with negative expression of MHC-I in the resulting cells, with highest ablation rate seen in sgB2M1 (Fig 1). In the subsequent experiments, we have used this gRNA to ensure adequate sensitivity of the experiments. The other less efficacious sgRNAs against *B2M*, however, can be useful in situations where higher selection pressure is needed.

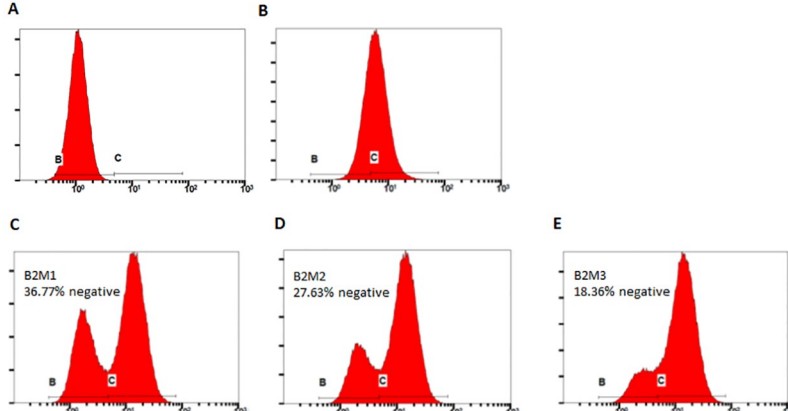

**Fig 1. Efficacy of different sgRNAs against *B2M* in ablating cell surface MHC-I antigen.** HEK293T cells were transfected with combination of pCMV-spCas9 and different sgRNAs against *B2M* (B2M1, C; B2M2, D; and B2M3, E), at a ratio of 1:1.5 for Cas9 and sgRNA plasmid. 5 days after transfection, they were stained with FITC-conjugated antibody against human HLA-A,B&C and run on flow cytometer. A and B were negative (no antibody staining) and positive (wild-type HEK293T cells stained with antibody) controls, respectively.

## Effective enrichment of indels and DKOs by FAME for both tumor suppressor genes and oncogenes

The proposed workflow for FAME is illustrated in Fig 2. On day 0, plasmids for expressing Cas9 and sgRNAs against *B2M* and gene-of-interest are transfected into target cells; if the sgRNA against gene-of-interest has not previously been tested, T7EI or Surveyor assay can be conducted 2 days after transfection to verify the efficacy of the sgRNA as an optional step. On day 5, the transfected cells are stained with anti-MHC-I antibody, and cells with surface MHC-I ablation can be selected with FACS, with the negative population being sorted into single cells in 96-well plates for monoclonal expansion. The presence and nature of the mutations

Day 0 – Transfection of target cells with plasmids expressing Cas9 and sgRNAs for B2M and gene-of-interest

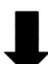

Day 2 – T7EI assay to confirm mutagenesis efficacy of selected sgRNAs

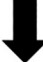

Day 5 – FACS of cells with surface MHC-I ablated into 96-well plate

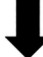

Expansion in culture

Day 15-20 – Verification of DKO by Sanger Sequencing

**Fig 2. Workflow for FAME.** See text for detailed explanation.

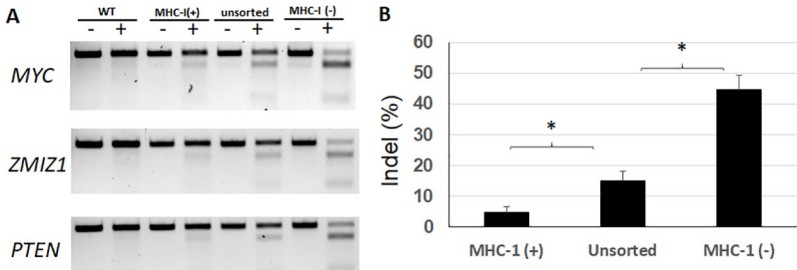

**Fig 3. Enrichment of *indels* with FAME strategy in diverse target genes.** A. HEK293T cells were transfected with expression plasmids for spCas9, sgRNAs for *B2M* and individual genes-of-interest (*MYC, ZMIZ1 and PTEN)*. 5 days later, transfected cells were stained with FITC-conjugated antibody against human HLA-A,B&C, sorted into pools of negative and positive cells, and expanded in culture. The pooled cells were amplified for genomic DNA extraction. For each T7EI assay, genomic DNA from 4 samples were included, i.e., wild-type HEK293T cells, MHC-I positive pool, unsorted pool, and MHC-I negative pool. For each sample, both undigested (-) and digested (+) PCR products by T7E1 were run for comparison and quantification. B. Comparison of indel frequencies for MHC-1 (+), unsorted, and MHC-1 (-) cell population. Indel frequencies were calculated using band signal intensity as reported by image analysis of the original gel pictures from the three T7EI assays represented in panel A. Student's t-test was used to compare the indel frequencies between MHC-1 (+) and unsorted cells, and between unsorted and MHC-1 (-) cells. * p<0.001.

in each colony are then determined by Sanger sequencing of the PCR products spanning the target sites either directly, or after cloning the PCR products into a plasmid. The entire process should take around 2–3 weeks, significantly reducing human labor and time spent on colony isolation and mutation verification.

To prove that such a scheme effectively enriches the frequency of indels in the target genes, we conducted three co-targeting experiments using the FAME strategy against *PTEN*, *MYC and ZMIZ1* in HEK293T cells. *PTEN* is a tumor suppressor gene [14], while *MYC* and *ZMIZ1* are both oncogenes [15, 16]. For each co-targeting experiment, 5 days after transfection, cells were sorted into MHC-I (-) and (+) populations after staining with anti-MHC-I antibody. These populations were expanded in culture, along with unsorted population. Genomic DNA from each pool was extracted and used for T7EI assay. The digestion product was run on agarose gel (Fig 3A), and the captured images were analyzed for estimation of indel frequency in each population of cells. For all three targeted genes, the indel frequencies were significantly suppressed in the MHC-I (+) populations (4.8%), compared to that of unsorted cells (15%), while highly enhanced in MHC-1 (-) cells (44.5%). All these changes compared to unsorted cells were highly statistically significant (p < 0.001) (Fig 3B).

High indel frequency approaching 50% implies that significant amount of MHC-I negative cells are DKOs, i.e., harboring biallelic mutations. In fact, at these levels, indel frequencies are likely underestimated for signal saturation [8]. These results bode well that we may be able to isolate DKO subclones from the MHC-I (-) populations with less efforts. To prove this, we sorted HEK293T cells co-targeted for *B2M and PTEN* with either negative or positive surface MHC-I into single cells that were seeded in 96-well plates (Fig 4A). We picked 9 clones each from MHC-I positive and negative populations, and PCR amplified the targeted loci with the same primers used for T7EI assay. We next sequenced the PCR products with the *PTEN* reverse primer. We were able to determine all 9 clones from MHC-I positive cells are unaltered at this locus (Fig 4B); on the strength of sequencing PCR products alone we were also able to determine 3 of the 9 MHC-I negative clones were DKOs (Fig 4C as an example); For the remaining 6 MHC-I negative clones with illegible sequencing results, we cloned the PCR products into a plasmid vector and sequenced resulting plasmids, and were finally able to decide all 9/9 MHC-I negative colonies were true DKOs. One example was shown in Fig 4D–4F. The

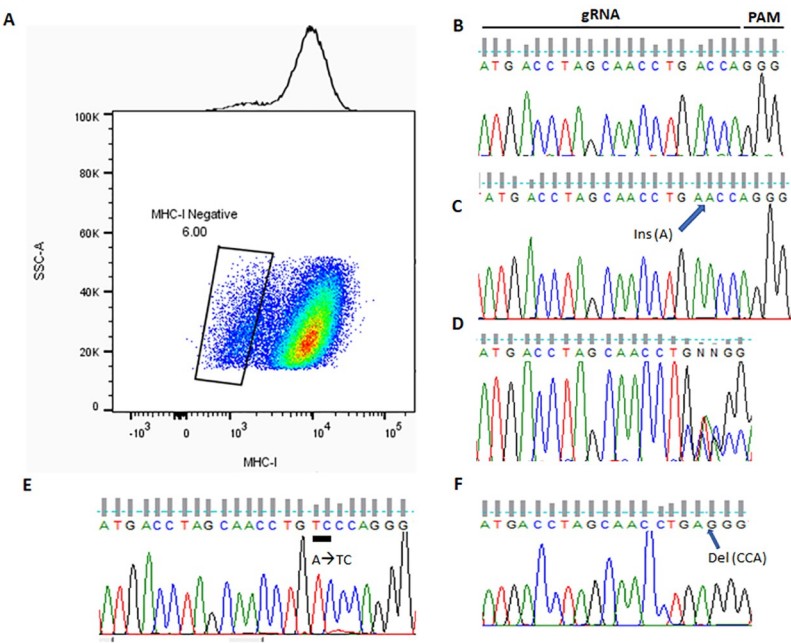

**Fig 4. Enrichment of *DKO*s with FAME strategy in HEK293T cells co-targeted for B2M and PTEN.** (A) HEK293T cells were transfected with expression plasmids for spCas9, and sgRNAs for *B2M and PTEN*. 5 days later, transfected cells were stained with FITC-conjugated antibody against human HLA-A,B&C. Single isolated cells with positive and negative surface MHC-I were sorted into 96-well plates. B-F. After expansion of the isolated clones, the region targeted by sgRNA was PCR amplified from genomic DNA, and submitted for Sanger sequencing. Representative chromograms were shown in the context of the target site, including PAM (GGG), for the picked clones. B, a typical wild type colony from MHC-I positive population identified with PCR sequencing; C, a typical DKO colony from MHC-I negative population with homozygous insertions identified with PCR sequencing; D, a colony (#4) from MHC-I negative population with illegible chromatogram with mixed signals; E and F, Sanger sequencing of plasmids with cloned PCR products from the same MHC-I negative colony as in D (#4) identified the exact mutations in each allele.

sequencing of the PCR product for this particular clone showed mutations in the targeted loci, but was not able to determine if they are in one allele or both (Fig 4D). After subcloning of the PCR products followed by Sanger sequencing, we were able to show that both alleles harbored different mutations (Fig 4E and 4F). The above results provide a solid proof-of-principle that FACS-based negative selection using membrane markers could be an effective method for enriching biallelic indel mutations in the target loci.

## Discussion

The technology of genome editing has evolved rapidly over the last decade, to the extent that it does not impose much technical challenges for routine use in any average biomedical laboratories with training in basic molecular and cell biology. The main hurdles to wider adoption likely stem from high cost in terms of time, labor and expenditure, which compare unfavorably to alternative technologies such as RNA interference. This is unfortunate as genome editing, especially with CRISPR, indeed has greater potential not only in basic research but also in clinical applications [17].

In this report, we provided proof-of-principle evidence that a negative selection scheme using surface marker could be useful in enriching indel mutations and DKOs in genome editing experiments. Our results have fulfilled the goals we set out to achieve, i.e., to improve the efficacy for obtaining DKO mutants and at the same time improve the workflow efficiency by

automation. We trust that our innovation reported here could help overcome existing technical hurdle for wider adoption of gene editing by all biomedical scientists with basic molecular and cell biology training and access to FACS equipment.

It is intriguing that the sequencing results from the 9 MHC-1 negative subclones of HEK293T cells co-targeted for *PTEN* revealed that a minority of them harbor homozygous mutations (Fig 3). As Cas9 induced DNA breaks on separate alleles should be independent of each other, the odds that both alleles develop the same mutation are extremely rare. Even though these homozygous biallelic mutations remain a minority, they could not be attributed to chance alone. We postulate that these are the results of sequential repair of the two alleles, with the first allele repaired by non-homologous end-joining, leading to indels, and the second by homology directed repair, using the now repaired first allele as a template, thus copying the exact mutation from the first allele to the second. Alternatively, a large deletion at the second allele may have occurred that have affected one of the primer binding sites, leading to lack of representation of this second allele in the PCR products. In either case, the conclusion of this experiment remains valid, and these subclones remain a minority.

We were able to achieve significant mutation enrichments for both tumor suppressor genes and oncogenes. It is reasonable to believe that our technology could be used for most genes except for those truly essential ones, for which DKO cells would not be able to survive without a rescue system in place. Our system was tested in human cells (HEK293T), and we have no illusion that our technology can be used in all other systems without adaptation. For example, new sgRNAs need to be designed and tested for homologs of *B2M* in cells from other species such as rodents. It is also well known that many cancer cells lose expression of MHC I antigen, as a way of immune escape [18], and our selection scheme would not work in such cells. For these reasons, it is imperative that we continue to search for additional membrane selectable markers that fit the criteria that we put forth earlier, i.e., selectable with FACS, autosomal, and non-essential for survival and other important functions.

With that said, *B2M* remains a desired target for generating hypoimmunogenic cell therapy products. With ablation of *B2M*, the exogenous cells no longer are target of rejection by cell immunity from the hosts, making low-cost and large-scale production of certain products from allogenic sources possible. The possibility was previously illustrated in xenograft of human ES cells, and we foresee that our technology will facilitate future development of off-shelf cell therapy products for many different diseases.

## Supporting information

**S1 Raw image.**
(TIF)

## Acknowledgments

We thank Dr. Branden Moriarity of University of Minnesota for providing key materials and reading the manuscript.

## Author Contributions

**Conceptualization:** Ming V. Li.

**Data curation:** Michael Hansen, Ming V. Li.

**Formal analysis:** Sara Bowen, Ming V. Li.

**Funding acquisition:** David A. Largaespada, Ming V. Li.

**Investigation:** Michael Hansen, Ming V. Li.

**Methodology:** Michael Hansen, Xiaopin Cai, Sara Bowen, Ming V. Li.

**Project administration:** Michael Hansen, Xiaopin Cai, Sara Bowen, Ming V. Li.

**Resources:** David A. Largaespada, Ming V. Li.

**Software:** Sara Bowen.

**Supervision:** David A. Largaespada, Ming V. Li.

**Validation:** Sara Bowen, Ming V. Li.

**Visualization:** Sara Bowen, Ming V. Li.

**Writing – original draft:** Ming V. Li.

**Writing – review & editing:** David A. Largaespada, Ming V. Li.

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
