## [Decision Letter · Decision Letter 0]

21 Jul 2020

PONE-D-20-17056

Flow Assisted Mutation Enrichment (FAME): A Highly Efficacious and Efficient Method to Enriching Double Knockouts (DKO) after Gene Editing

PLOS ONE

Dear Dr. Li,

Thank you for submitting your manuscript to PLOS ONE. After careful consideration, we feel that it has merit but does not fully meet PLOS ONE’s publication criteria as it currently stands. Therefore, we invite you to submit a revised version of the manuscript that addresses the points raised during the review process.

Please provide statistical analysis of your results as required by *PLoS One*. Please provide sufficient detail of your methods that they can be replicated by others. Please try to replicate your results in another diploid cell line or in iPS cells.

We look forward to receiving your revised manuscript.

Kind regards,

Alfred S Lewin, Ph.D.

Academic Editor

PLOS ONE

Journal Requirements:

2. We note that you have a patent relating to material pertinent to this article. Please provide an amended statement of Competing Interests to declare this patent (with details including name and number), along with any other relevant declarations relating to employment, consultancy, patents, products in development or modified products etc. Please confirm that this does not alter your adherence to all PLOS ONE policies on sharing data and materials, as detailed online in our guide for authors http://journals.plos.org/plosone/s/competing-interests by including the following statement: "This does not alter our adherence to  PLOS ONE policies on sharing data and materials.” If there are restrictions on sharing of data and/or materials, please state these. Please note that we cannot proceed with consideration of your article until this information has been declared.

4. Please ensure that you refer to Figure 4 in your text as, if accepted, production will need this reference to link the reader to the figure.

Reviewers' comments:

Reviewer's Responses to Questions

**Comments to the Author**

1. Is the manuscript technically sound, and do the data support the conclusions?

Reviewer #1: Yes

Reviewer #2: Yes

2. Has the statistical analysis been performed appropriately and rigorously? 

Reviewer #1: N/A

Reviewer #2: N/A

3. Have the authors made all data underlying the findings in their manuscript fully available?

Reviewer #1: No

Reviewer #2: No

4. Is the manuscript presented in an intelligible fashion and written in standard English?

Reviewer #1: Yes

Reviewer #2: Yes

5. Review Comments to the Author

Reviewer #1: The works is scientifically clear and experiments have been performed correctly.

However, as I highlighted in my review no statistic were included to the figures and also is not clear how many times experiments have been repeated.

The fact that three different locus show similar results highly support the robustness of the method but it's not clear how many time per locus the experiments have been performed.

The authors should better explain and address this.

Reviewer #2: The manuscript reports a very promising method for easier isolation of cells successfully modified by gene editing with the CRISPR/Cas9 system. The data and method need to be better documented, as detailed below, before publication. The manuscript would also be significantly strengthened if the authors validated their method in a second cell line (for example diploid RPE1 or iPS cells and not only in HEK293T cells) as well as using a second method for Cas9 expression (for example using Cas9 RNP rather than expression plasmids).

1) It is essential that authors quantify the enrichment achieved by B2M negative selection. The frequency of mutations at the targeted sites was determined by the T7E1 assay. The %modification can easily be calculated (for example as detailed in the protocol from NEB) and used to derive the enrichment provided by selection for B2M-negative cells. This information will be useful to better document the efficiency of the co-selection method proposed here.

However, it is also well known that the T7E1 assay does not always give a very accurate quantification of %modification (in particular for high rates of modification) and other simple and more accurate methods can be used such as TIDE or ICE, which are based on analysis of chromatograms resulting from Sanger sequencing of PCR products of the target locus.

2) Some key experimental details are missing from the methods section. It is essential to provide the missing information so that other researchers can use the method.

- provide the amounts of plasmid transfected (specifically including the relative amounts of plasmid for B2M and target gene guide RNAs)

- indicate how guide RNA sequences were selected (in particular,which software was used and how off-targets were minimized).

- provide the reference and source of MHC-I antibody used for FACS

3) Were several guide RNAs tested for target genes PTEN, MYC and ZMIZ1? The T7E1 assays shown in Figures 3 and 4 suggest that the guides used in co-selection experiments are relatively inefficient. Is the enrichment achieved by B2M negative selection comparable for guides of different efficiencies? ….and specifically for guides of higher efficiency?

3) Even though they were mostly developed to enrich for HDR-mediated gene editing, previously published selection methods should be more extensively cited in the introduction. In particular:

Nat Methods. 2017 Jun;14(6):615-620 (which shows enrichment for both NHEJ- and HDR-mediated gene editing).

Nucleic Acids Res. 44, 7997–8010 (2016)

Methods. 2017 May 15;121-122:45-54.

Minor comments

1) The authors used the commercial kit from NEB, called ENGEN. Please use the generic name of the T7E1 assay (which was developed long before the kit) rather than the NEB name in the main text (for example line 44, 195, etc…) and in Figure 2.

2) The authors discuss enrichment for tumor suppressor genes and oncogenes. It is unclear if the tumor suppressor gene and oncogenes selected for their proof of principle experiments are known to impact proliferation of HEK293T cells. Such information, if available, should be provided. If that information is not known, the authors should simply introduce the target genes as examples for testing their method.

3) Sequencing analysis (Figure 3) showed that many clones only contained a single mutant sequence. Since there are at least two copies of the B2M gene in HEK293T cells, finding a single mutant sequence is unexpected and should be commented in the text. One possibility is that only one allele could be amplified by PCR and that the second allele could not be amplified because a large deletion, including at least one primer sequence, had taken place. This possibility could be investigated by performing PCR with primers located further away from the target sequence.

6. PLOS authors have the option to publish the peer review history of their article (what does this mean?). If published, this will include your full peer review and any attached files.

Reviewer #1: No

Reviewer #2: No

---

## [Author Response · Author response to Decision Letter 0]

21 Jan 2021

Response to Dr. Levin’s editorial decision comments:

Our Response: Thanks for the kind reminder on the style requirements. I have read carefully the above instructions and trust the revised manuscript now meets these requirements. 

2. We note that you have a patent relating to material pertinent to this article. Please provide an amended statement of Competing Interests to declare this patent (with details including name and number), along with any other relevant declarations relating to employment, consultancy, patents, products in development or modified products etc. Please confirm that this does not alter your adherence to all PLOS ONE policies on sharing data and materials, as detailed online in our guide for authors 

http://journals.plos.org/plosone/s/competing-interests

by including the following statement: "This does not alter our adherence to PLOS ONE policies on sharing data and materials.” If there are restrictions on sharing of data and/or materials, please state these. Please note that we cannot proceed with consideration of your article until this information has been declared.

Our Response: Thanks for clarification on the journal’s policy on needed declaration. As instructed, I have included a statement in the cover letter for this resubmission with the detailed information on the patent application and clarification that this does not alter our adherence to all PLOS ONE policies on sharing materials and data. In addition, we have in previous submission declared no competing interests on the part of all authors, and this remains true for this revision. 

In your cover letter, please note whether your blot/gel image data are in Supporting Information or posted at a public data repository, provide the repository URL if relevant, and provide specific details as to which raw blot/gel images, if any, are not available. Email us at plosone@plos.org if you have any questions

Our Response: Thanks again for clarification. In this revision, we are submitting all original uncropped and unadjusted images in the Supporting Information as instructed. This is noted in the cover letter as well.

4. Please ensure that you refer to Figure 4 in your text as, if accepted, production will need this reference to link the reader to the figure.

Our Response: Thank you for pointing out this apparent oversight, for which I apologize. In this current revision we made sure that all figures are referred to in the text. 

Responses to Reviewers’ comment

5. Review Comments to the Author

Reviewer #1: The works is scientifically clear and experiments have been performed correctly.

However, as I highlighted in my review no statistic were included to the figures and also is not clear how many times experiments have been repeated.

The fact that three different locus show similar results highly support the robustness of the method but it's not clear how many time per locus the experiments have been performed.

The authors should better explain and address this.

Our Response: We thank the reviewer for the kind comments and agree with the reviewer’s concern for lack of statistical analysis in the earlier manuscript. For each locus, we have done sorting once, but have repeated the experiments in three different loci. For each locus, we have also repeated the T7E1 assays (previously referred to as ENGEN assay in the first version of the manuscript) a few times, with all results showing the same pattern. High-quality representative images were selected in the presentation and also for image analysis for indel estimation. The three repeated experiments with highly consistent results allowed us to conduct statistical analysis after quantifying the band strength in T7E1 assays, which showed that the method is highly effective in enriching mutations, with a p value < 0.001 in relevant comparisons. 

Reviewer #2: The manuscript reports a very promising method for easier isolation of cells successfully modified by gene editing with the CRISPR/Cas9 system. The data and method need to be better documented, as detailed below, before publication. The manuscript would also be significantly strengthened if the authors validated their method in a second cell line (for example diploid RPE1 or iPS cells and not only in HEK293T cells) as well as using a second method for Cas9 expression (for example using Cas9 RNP rather than expression plasmids).

Our response: We thank the reviewer for the kind encouragement of our work and appreciate very much all suggestions by the reviewer and have been trying our best to follow them (see our point-to-point responses below). The article will certainly be strengthened by validity in a second cell line and/or an alternative method for introduction Cas9. At the time of initial submission, we only intend to introduce a proof of concept for using a neutral membrane protein as a marker for negative selection, which we trust that we have accomplished. We have been careful not to overclaim and in the discussion freely admit that this method is not an answer to all other types of cells without modifications, such as rodent cells and cancer cell lines that have already silenced MHC-I. After reading the reviewer’s suggestion, it was our intention to validate this in a second cell line, but as indicated earlier, the pandemic and my other responsibilities have stayed our hands, and we will not be able to pursue this in a timely fashion. Therefore, at this time, we will again limit the scope of the manuscript to a proof of a concept, while we are open to further development of this concept in future studies, when opportunities arise. 

1) It is essential that authors quantify the enrichment achieved by B2M negative selection. The frequency of mutations at the targeted sites was determined by the T7E1 assay. The %modification can easily be calculated (for example as detailed in the protocol from NEB) and used to derive the enrichment provided by selection for B2M-negative cells. This information will be useful to better document the efficiency of the co-selection method proposed here.

However, it is also well known that the T7E1 assay does not always give a very accurate quantification of %modification (in particular for high rates of modification) and other simple and more accurate methods can be used such as TIDE or ICE, which are based on analysis of chromatograms resulting from Sanger sequencing of PCR products of the target locus.

Our response: We thank the reviewer for the suggestion to quantify mutation enrichment and completely agree. We were able to quantify the enrichment based on the existing data from the T7E1 assay. Given the robustness of the enrichment with negative selection, the T7E1 assay has been adequate to prove that the enrichment is highly statistically significant (p < 0.001). The reviewer is spot-on regarding the accuracy of the T7EI assay in estimating the frequency of the indel, especially in the highly enriched MHC-I (-) population where indel% was pushed to close to 50%, a level indicating saturation and the formula for calculating indel% may significantly be underestimated. We admire the reviewer’s deep insight, and in hindsight, we wished that we have had this in mind when designing these experiments. We regret that we do not have existing data for more accurate analysis, and are unable to conduct them at this time, given the circumstances. We take comfort in the fact that the main conclusion that indel frequency was greatly improved by the sorting procedure remain intact.

2) Some key experimental details are missing from the methods section. It is essential to provide the missing information so that other researchers can use the method.

Our response: Thanks to the reviewer for the reminder to us to provide experimental details so that readers may reproduce these results. We have inserted comments in the methods section to answer the above questions and will also provide them below as well.

- provide the amounts of plasmid transfected (specifically including the relative amounts of plasmid for B2M and target gene guide RNAs)

Our response: We have included the following clarification in the transfection method: In a typical gene editing experiment with B2M negative selection, A total of 2.5 mcg of plasmid DNA (1 mcg pCMV-hCas9, 0.5 mcg pENTR221-U6-sgB2M1, and 1 mcg of plasmid for gRNA targeting the gene-of-interest) is used to transfect 1 well of HEK293T cells in a 6-well plate. 

- indicate how guide RNA sequences were selected (in particular, which software was used and how off-targets were minimized).

Our response: The software mentioned in the methods section under plasmid and cloning was Deskgen, a UK-based online algorithm for generating candidate gRNAs which unfortunately seemed to have ceased to exist at this time. It was chosen over other cloud-based designers at that time for convenience, as it provided both on-target and off-target scores as selection criteria. For each locus, we have chosen sgRNA with best scores on both on- and off-target effects. 

- provide the reference and source of MHC-I antibody used for FACS

Our response: the source of the MHC-I antibody is Biolegend (San Diego, CA), as listed under Flow Cytometry in the Methods section. The reference for this antibody was also cited in the Results section (reference #9)

3) Were several guide RNAs tested for target genes PTEN, MYC and ZMIZ1? The T7E1 assays shown in Figures 3 and 4 suggest that the guides used in co-selection experiments are relatively inefficient. Is the enrichment achieved by B2M negative selection comparable for guides of different efficiencies? ….and specifically for guides of higher efficiency?

Our response: Thanks to the reviewer for this important observation, with implications regarding if the enrichment method would be generally applicable for gRNAs with different levels of mutagenic strength. We have only chosen the highest scored gRNA for each locus based on the on- and off-target scores provided by Deskgen. The indel frequency calculated from the T7E1 assay from unsorted cells is on average 15% (Fig 3B), which is modest, but we think this resulted from gRNA against gene-of-interest being diluted by gRNAs against B2M. The indel frequency was further suppressed in MHC-1 (+) population to 4.8%, but this is expected. In addition, there could certainly be room for further optimization such as the amount of DNA, lipofectamine and the ratio for plasmids of Cas9, and guides for B2M and gene-of-interest. In the future, these parameters can be tested in detail, but for the time being are beyond this manuscript’s modest scope to provide a proof of concept. 

3) Even though they were mostly developed to enrich for HDR-mediated gene editing, previously published selection methods should be more extensively cited in the introduction. In particular:

Nat Methods. 2017 Jun;14(6):615-620 (which shows enrichment for both NHEJ- and HDR-mediated gene editing).

Nucleic Acids Res. 44, 7997–8010 (2016)

Methods. 2017 May 15;121-122:45-54.

Our response: Thanks to the reviewer for the suggestions to include additional previous efforts for mutation enrichment that have been influential in the field. As an former outsider to the gene editing field (I am a student of signal transduction and glucose metabolism), I am greatly enlightened by these readings and humbled by the depth of this exciting and rapidly evolving field. I am honored to be able to contribute to it with our own methods, which does appear to add to the tool kits of molecular genetics with its own advantages and shortcomings. We have since included these references in the introduction section. 

Minor comments

1) The authors used the commercial kit from NEB, called ENGEN. Please use the generic name of the T7E1 assay (which was developed long before the kit) rather than the NEB name in the main text (for example line 44, 195, etc…) and in Figure 2.

Our response: Thanks to the reviewer on the suggestion of using a generic name for the assay, which we completely agree. We have followed the advice through the entire text and figures. 

2) The authors discuss enrichment for tumor suppressor genes and oncogenes. It is unclear if the tumor suppressor gene and oncogenes selected for their proof of principle experiments are known to impact proliferation of HEK293T cells. Such information, if available, should be provided. If that information is not known, the authors should simply introduce the target genes as examples for testing their method.

Our response: We thank reviewer for the reminder to us to refrain from speculation and speak from evidence. This is a much appreciated and needed advice. We have reorganized the results so that all three targets are presented as genes at different end of functional spectrum without speculating their potential effects on cell growth of HEK293T, which we do not have any actual evidence, like the reviewer has pointed out. Such reorganization did not change the conclusion of the manuscript but has much improved its scientific integrity.

3) Sequencing analysis (Figure 3) showed that many clones only contained a single mutant sequence. Since there are at least two copies of the B2M gene in HEK293T cells, finding a single mutant sequence is unexpected and should be commented in the text. One possibility is that only one allele could be amplified by PCR and that the second allele could not be amplified because a large deletion, including at least one primer sequence, had taken place. This possibility could be investigated by performing PCR with primers located further away from the target sequence.

Our response: We very much appreciate the reviewer's insight for this unexpected finding. We were also intrigued by this phenomenon that a minority of the subclones (3 out of 9) seem to have acquired homozygous point mutations on both alleles based on the sequencing results of the genomic PCR. The reviewer has provided one valid explanation that a large deletion might have occurred, rendering sequencing of that variant not possible, and the suggestion of new set of PCR with primers further away from the target will be able to prove this hypothesis. 

These comments are much appreciated; we have our own theory as well, which goes that one allele may have been repaired first, leading to a point mutation by non-homologous end-joining. Subsequently, this now repaired first allele then becomes the repair template for homology directed repair for the second strand, leading to reproduction of the exact same mutation on the second allele. In either case, the conclusion that these minority subclones have biallelic mutations remain valid. We agreed with the reviewers that this unexpected finding needs to be discussed and have covered this with a new paragraph in the discussion section. As much as we want to test these theories, we are currently not able to conduct wet experiments for reasons explained earlier. We do however believe that the main conclusion of the manuscript is still intact, and thankfully, these clones are only a minority.

---

## [Decision Letter · Decision Letter 1]

8 Feb 2021

Flow assisted mutation enrichment (FAME): a highly efficacious and efficient method to enrich double knockouts (DKO) after gene editing

PONE-D-20-17056R1

Dear Dr. Li,

We’re pleased to inform you that your manuscript has been judged scientifically suitable for publication and will be formally accepted for publication once it meets all outstanding technical requirements.

Kind regards,

Alfred S Lewin, Ph.D.

Section Editor

PLOS ONE

Additional Editor Comments (optional):

Reviewers' comments:

Reviewer's Responses to Questions

**Comments to the Author**

1. If the authors have adequately addressed your comments raised in a previous round of review and you feel that this manuscript is now acceptable for publication, you may indicate that here to bypass the “Comments to the Author” section, enter your conflict of interest statement in the “Confidential to Editor” section, and submit your "Accept" recommendation.

Reviewer #1: All comments have been addressed

Reviewer #2: All comments have been addressed

2. Is the manuscript technically sound, and do the data support the conclusions?

Reviewer #1: Yes

Reviewer #2: Yes

3. Has the statistical analysis been performed appropriately and rigorously? 

Reviewer #1: Yes

Reviewer #2: N/A

4. Have the authors made all data underlying the findings in their manuscript fully available?

Reviewer #1: Yes

Reviewer #2: Yes

5. Is the manuscript presented in an intelligible fashion and written in standard English?

Reviewer #1: Yes

Reviewer #2: Yes

6. Review Comments to the Author

Reviewer #1: (No Response)

Reviewer #2: The authors have adressed my main concerns relative to the text and included imoprtant technical information requested to ensure the technique can be implemented by other researchers.

7. PLOS authors have the option to publish the peer review history of their article (what does this mean?). If published, this will include your full peer review and any attached files.

Reviewer #1: No

Reviewer #2: No

---

## [Editor Report · Acceptance letter]

23 Feb 2021

PONE-D-20-17056R1 

Flow assisted mutation enrichment (FAME): a highly efficacious and efficient method to enrich double knockouts (DKO) after gene editing 

Dear Dr. Li:

I'm pleased to inform you that your manuscript has been deemed suitable for publication in PLOS ONE. Congratulations! Your manuscript is now with our production department. 

Kind regards, 

on behalf of

Dr. Alfred S Lewin 

Section Editor

PLOS ONE